# Energy exchanges at contact events guide sensorimotor integration

Ali Farshchian[1,2]*, Alessandra Sciutti[2,3], Assaf Pressman[4], Ilana Nisky[4,5], Ferdinando A Mussa-Ivaldi[1,2,6,7]

[1]Department of Biomedical Engineering, Northwestern University, Evanston, United States; [2]Sensory Motor Performance Program, Rehabilitation Institute of Chicago, Chicago, United States; [3]Department of Robotics, Brain and Cognitive Sciences, Italian Institute of Technology, Genoa, Italy; [4]Department of Biomedical Engineering, Ben-Gurion University of the Negev, Beersheba, Israel; [5]Zlotowski Center for Neuroscience, Ben-Gurion University of the Negev, Beersheba, Israel; [6]Department of Physical Medicine and Rehabilitation, Northwestern University, Chicago, United States; [7]Department of Physiology, Northwestern University, Chicago, United States

**Abstract** The brain must consider the arm's inertia to predict the arm's movements elicited by commands impressed upon the muscles. Here, we present evidence suggesting that the integration of sensory information leading to the representation of the arm's inertia does not take place continuously in time but only at discrete transient events, in which kinetic energy is exchanged between the arm and the environment. We used a visuomotor delay to induce cross-modal variations in state feedback and uncovered that the difference between visual and proprioceptive velocity estimations at isolated collision events was compensated by a change in the representation of arm inertia. The compensation maintained an invariant estimate across modalities of the expected energy exchange with the environment. This invariance captures different types of dysmetria observed across individuals following prolonged exposure to a fixed intermodal temporal perturbation and provides a new interpretation for cerebellar ataxia.
DOI: https://doi.org/10.7554/eLife.32587.001

*For correspondence: a-farshchiansadegh@ northwestern.edu

## Introduction

In a conference if you cannot understand the speaker due to excessive background noise or poor acoustics, seeing her face would help you capture what she is saying. The evident explanation for this experience is that the integration of information from multiple sensory modalities improves perception (*Ernst and Bülthoff, 2004*). Similarly, the sensorimotor control system combines different sensory measurements to enhance the perception required to perform accurate movements and to skillfully manipulate objects. However, because of delays in neural pathways, the brain cannot rely entirely on sensory feedback to effectively control movements, particularly when interacting with a dynamical environment. Predicting the consequences of an action is essential to compensate for the temporal delays of sensory information. To this end, a widely accepted view is that the brain relies on internal representations, or 'internal models' of the body and of the environment in which it operates (*Wolpert et al., 1995*; *Wolpert and Miall, 1996*; *Wolpert and Kawato, 1998*; *Kawato, 1999*). The predictions of these internal models, often called forward models, generate expectations for future sensory consequences of the ongoing motor commands before sensory feedback becomes available (*Shadmehr et al., 2010*). These 'priors' are combined with delayed sensory feedback to estimate both the state (e.g. position and velocity) of the body and the context (e.g. mass of

manipulated object) of the movement (*Wolpert and Ghahramani, 2000*; *Wolpert and Flanagan, 2001*). In a biological system, however, noise and uncertainty spread through every aspect of sensory perception and motor command generation (*Faisal et al., 2008*). Additionally, the environment itself is ambiguous and variable. This makes state and context estimation probabilistic problems to solve. Over the past decade, Bayesian integration theory has provided a unifying framework to capture behavior under uncertainty in a wide range of psychophysical studies on sensory perception (*Weiss et al., 2002*; *Jazayeri and Shadlen, 2010*), multisensory integration (*Ernst and Banks, 2002*; *Alais and Burr, 2004*; *Ernst, 2007*), and sensorimotor function (*Körding and Wolpert, 2004*; *Miyazaki et al., 2005*). However, the temporal structure of state and context estimation remains largely unknown.

Object manipulation is an effective and natural test bed for sensorimotor integration. It engages multiple sensory modalities and in contrast to movements in free space, it provides an additional challenge to the nervous system. Holding an object changes the dynamics of the arm, thereby successful manipulation requires not only knowledge of the arm dynamics, but also knowledge of the object dynamics. This knowledge is not solely acquired through proprioceptive and tactile feedback; vision also provides information about the mechanical properties of the object (*Gilden and Proffitt, 1989*; *Gordon et al., 1991*; *Jenmalm and Johansson, 1997*; *Salimi et al., 2003*; *Ingram et al., 2010*; *Takamuku and Gomi, 2015*). Here we employed an object manipulation task to investigate the temporal resolution of the sensory integration process that provides the information for estimating the mechanical properties of the object being manipulated (i.e. context estimation). We considered two possibilities: a time-dependent structure in which context estimation takes place continuously or periodically at isochronous intervals and a state-dependent structure in which context estimation occurs sporadically at salient task-relevant events (e.g. contact events in an object manipulation task).

To test these alternative possibilities, we developed a virtual two-dimensional ping-pong game in which participants continuously manipulated an object (paddle) to hit a ball (*Figure 1A*). Visual, haptic, and auditory feedbacks were provided simultaneously (within the resolution and synchronization capabilities of our setup) at the time of impact between the paddle and the ball. This design was ideal for our purpose as it was a continuous object manipulation task that also included discrete multisensory events. In this task, the two proposed temporal structures would provide different mass estimations after adaptation to an artificial delay in the sensory feedback (*Foulkes and Miall, 2000*; *Miall and Jackson, 2006*; *Farshchiansadegh et al., 2015*). *Figure 1B* is a schematic illustration of the changes in the hand position in a reciprocal movement in which one hits the ball and returns back in preparation for the next hit in the pong game with its delayed visual representation. If proprioceptive and visual information are integrated continuously or periodically to estimate the mass of the paddle, then the internal representation of the mass should remain unchanged at the end of adaptation. This is because the mismatch between the two sensory measurements would integrate to zero (integrating over the region indicated by the gray box in *Figure 1B*) not only for position, but also for all the higher derivatives. On the other hand, if sensory integration for mass estimation occurs only at collision events, because collisions only occur when the hand is moving in the outward direction and therefore the sensory discrepancies do not integrate to zero across collision events, this should result in predictable and systematic changes in the mass representation depending on the difference between sensory measurements at the time of events. To assess the changes in representation of mass, we asked participants to perform reaching movements without feedback (in a feedforward fashion) before and after playing pong.

## Results

We asked three groups of volunteers to make blind reaching movements to visual targets before and after playing a simulated pong game holding a robotic manipulandum. After playing pong for a few minutes without a delay, the game's response to the player's movements was delayed and participants continued playing for ~40 min. We investigated the effects of adaptation on the reaching trajectories.

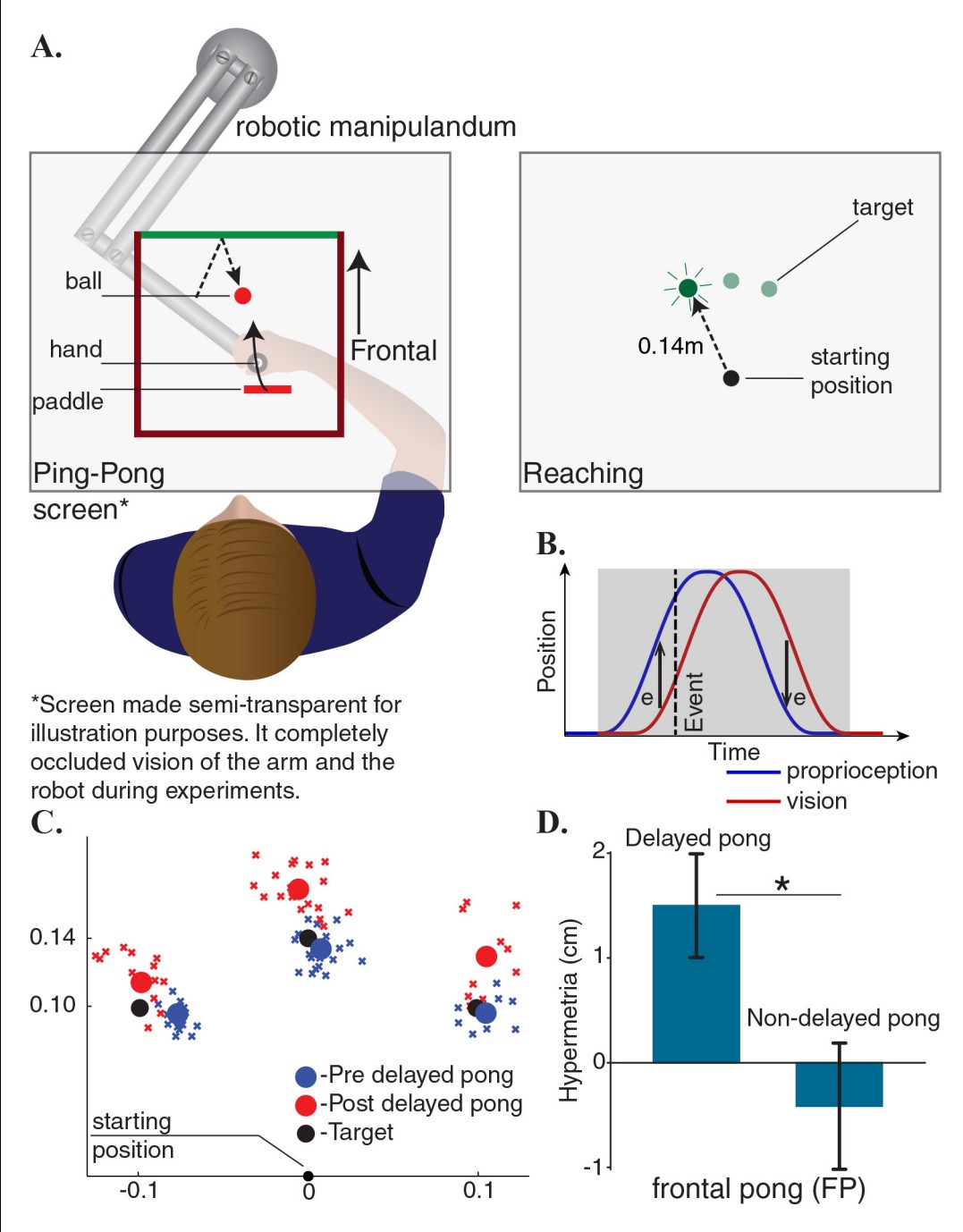

**Figure 1.** Adaptation to delayed feedback in a ping-pong game influences reaching behavior. (**A**) Subjects played a planar pong game in frontal direction using a robotic manipulandum. In addition to continuous visual feedback, auditory and tactile feedbacks were provided simultaneously upon collisions with the ball. After few minutes of familiarization, the game's response to the player's movements was delayed and subjects continued playing the game in the delayed environment. Participants also performed reaching movements without any continuous or terminal feedback before and after playing the pong. Objects and labels in black were not visible to the subjects. (**B**) A cartoon of the changes in the hand position during a reciprocal movement in the pong game and its delayed representation. If sensory integration occurs continuously, then the reaching trajectories should remain unchanged after adaptation because the difference between visual and proprioceptive information integrates to zero. However, if sensory integration occurs only at collisions, this should result in predictable changes in the terminal position of the reaching movements depending on the sensory measurements at collisions. (**C**) The endpoints of the reaching movements of a typical subject before and after adaptation. (**D**) All subjects showed hypermetria in

*Figure 1 continued on next page*

*Figure 1 continued*

the reaching movements after adaption. The hypermetria was absent in a subgroup who additionally did the same experiment without the delay. Error bars represent one standard error of the mean.

DOI: https://doi.org/10.7554/eLife.32587.002

## Experiment I

The first group of participants played a frontal pong (FP, proximal-distal direction, *Figure 1A*). With practice, all subjects improved their performance. Since subjects were instructed to maximize the number of collisions with the ball, hit rate was set as a metric for proficiency. A paired t-test between the first and the last five minutes of the delayed pong revealed a significant increase in the number of hits per minute ($p = 0.04$). Notably, playing the delayed pong influenced the reaching behavior. *Figure 1C* compares the endpoint of the reaches of a participant in this group before and after adaptation. A systematic hypermetria in reaching was observed in all subjects after playing the game (*Figure 1D*). The magnitude of the movements was significantly larger following adaptation (paired t-test, $p = 0.02$). To further verify that the changes in reaching trajectories were not a byproduct of interacting with the robot itself, a subgroup of the subjects in this group also participated in a control experiment in which the game was not delayed. Expectedly, the hypermetria was absent in this experiment (paired t-test, $p = 0.53$).

One interpretation of these results (our hypothesis) would suggest that adapting to the delay changed the representation of the mass of the object (paddle) being manipulated. In this case, hypermetria would follow from assigning inertial values to the object that are higher than the actual value. However, there were multiple alternative interpretations including different kinematic models (see the end of the Results section) that were similarly successful to explain this outcome. To consider these alternative explanations, we designed additional experiments in which participants played the pong game in lateral direction. The main objective of the lateral pong was to create a scenario in which two groups play the game under similar kinematic conditions but with paddles that possess different mechanical properties. This setup allowed us to tease apart the relative importance of kinematic and dynamic parameters that influence adaptation. To this end, we took advantage of the passive dynamics of the robot and asked two groups of participants to play pong in different regions of the workspace of the robot. The anisotropic position-dependent inertial properties of the robot effectively made the dynamics of the paddle to be different between the two groups. In this scenario if the adaptation is derived by the kinematic features of the pong game then the post adaptation effects on the reaching trajectories should be symmetric between the two groups. However, if adaptation is dominated by the dynamic features, then this should lead to asymmetric results.

## Experiments II and III

In these experiments, we placed two pong courts next to each other and participants played a lateral pong (LP, medio-lateral direction, *Figure 2A*). One group played the delayed pong only in the right court ($LP_R$), while the other group played the delayed pong only in the left court ($LP_L$). The same pattern of reach targets that was utilized in the experiment I were re-positioned within each court (*Figure 2A*). Both groups performed blind reaching movements to all six targets from the corresponding starting positions in each side before and after adaptation. To ensure that the difficulty level of playing pong was not different between the courts, initially all participants played the game with no delay in both courts. Hit rate analysis showed that there was no difference in performance across the courts (paired t-test, $p = 0.32$). Thus, we could assume that there was not an inherent gap in difficulty between the two courts. In addition, there was no significant difference in the movement extent (t-test, $p = 0.5$) between the movements made by the $LP_R$ group on the right court and the movements made by the $LP_L$ group on the left court during the pre-adaptation pong. Task performance was drastically affected when the delay was introduced. However, with practice both groups improved their performance significantly at an equivalent level. A mixed-design ANOVA with practice as a within-subject factor (2 levels) and group as a between-subject factor (2 levels) revealed a main effect of practice ($F(1,14) = 55, p<0.001$), no effect of group ($F(1,14) = 0.007, p = 0.93$) and no interaction effect ($F(1,14) = 2.1, p = 0.17$). Furthermore, there was no difference in movement extent (t-test, $p = 0.18$) between the two groups at the end of the adaptation.

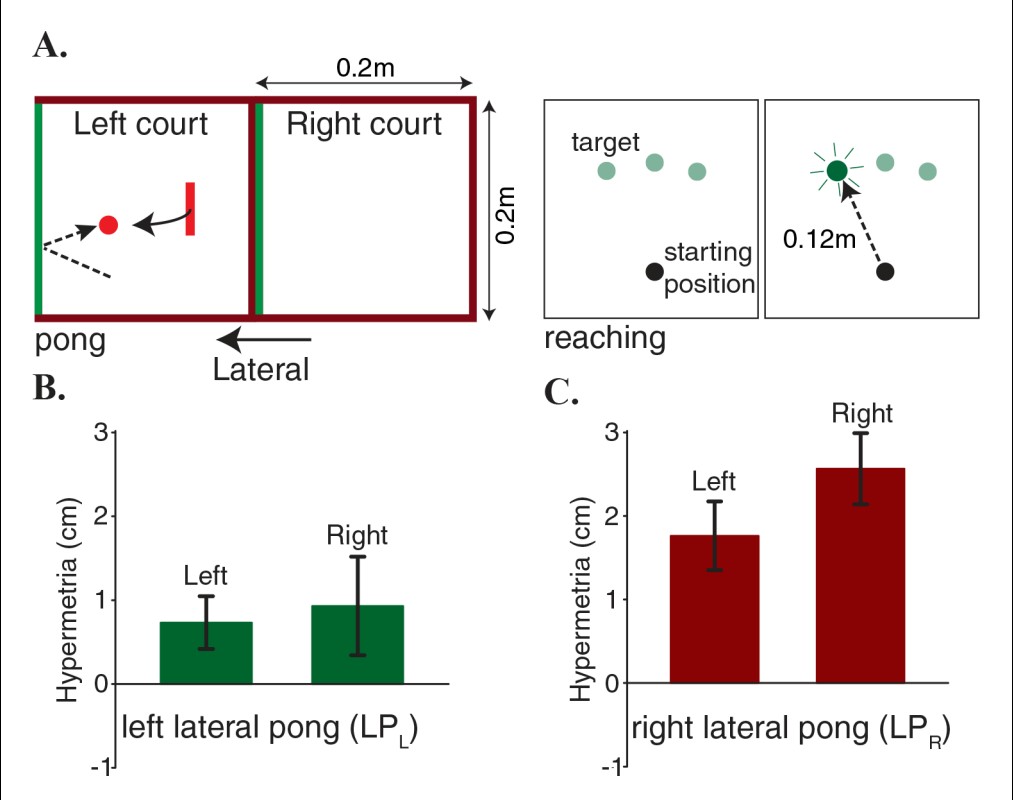

**Figure 2.** Hypermetria in reaching depends on the dynamics of the pong. (A) Two separate groups of subjects played a lateral pong with delay. Each group adapted to the delay only in one of the two courts that were placed next to each other. Both groups performed blind reaching movements before and after adaption from starting positions on the two sides of the body midline. Objects and labels in black were not visible to the subjects. (B) Subjects on the left court showed a very small average hypermetria on both sides. (C) Subjects on the right court showed a large average hypermetria on the right side that generalized to a lesser extent to the left side. Error bars represent one standard error of the mean.
DOI: https://doi.org/10.7554/eLife.32587.003

While the learning rates and the level of performance were largely equivalent across the two groups, the effect of adaptation on the reaching trajectories was strikingly different: the $LP_R$ group demonstrated a large hypermetria on the right side (the training region) that generalized to a lesser extent to the other side (*Figure 2C*), whereas the $LP_L$ group showed only a very small hypermetria on both sides (*Figure 2B*). A two-way mixed ANOVA on change in the movement magnitude, with reaching side as a within-subject factor (2 levels) and group as a between-subject factor (2 levels) revealed no significant main effect of reaching side ($F_{(1, 14)} = 1.2, p = 0.3$). However, there was a significant main effect of group ($F_{(1, 14)} = 5.6, p = 0.03$). Additionally, there was no significant inter-action effect ($F_{(1, 14)} = 3.2, p = 0.1$).

## Sensory integration at events explains individual differences

We have recently shown that when transporting an object carried by the hand, visual and proprioceptive information are integrated to optimize the kinetic energy transferred to the object (*Farshchiansadegh et al., 2016*). For the same optimization to occur in a pong game, it is necessary for the collisions to happen at the time of peak paddle velocity. Analysis on the relationship between the velocity profile and the collision time in the baseline non-delayed trials - when vision and proprioception were congruent - reveals that, here as well, participants adopted the energy-efficient strategy by hitting the ball, on average, at the time of peak velocity (*Figure 3A*). Introducing the delay affected this optimal behavior but participants exhibited a continuous effort towards recovering the energy optimal behavior. We have computed the time difference between the average time of

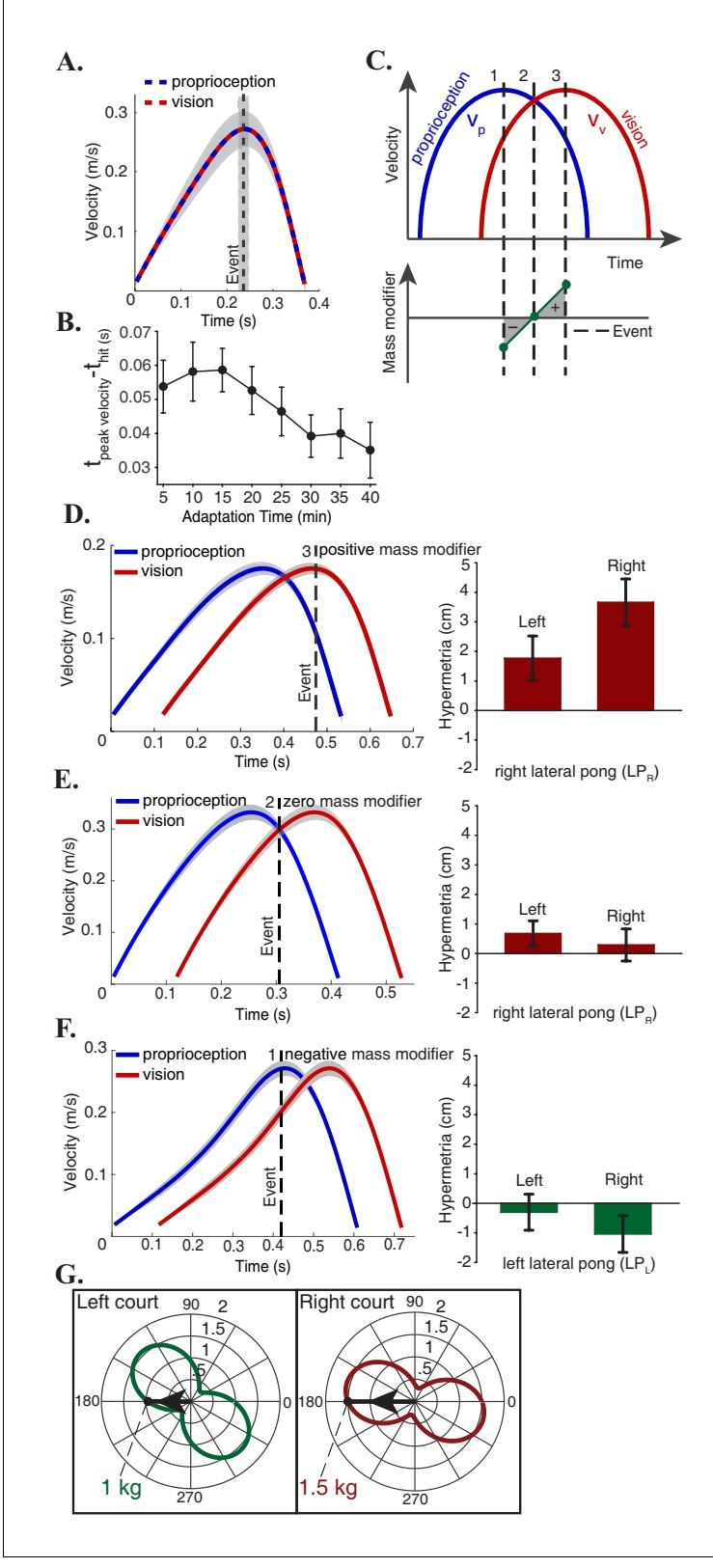

**Figure 3.** Mass modifier explains individual and group differences. (A) Subjects optimized the energetic cost of their movements in the non-delayed pong game by hitting the ball at the peak velocity of the paddle. (B) In adaptation trials, there was a continuous effort towards recovering the energy optimal policy by reducing the difference between the time of the peak velocity of the paddle and the time of the impact (C) In the delayed

*Figure 3 continued on next page*

*Figure 3 continued*

pong, visual and proprioceptive measurements were different at the time of collisions. Hence, sensory integration at events caused a misperception of the paddle's mass. Mass modifier is the difference between the actual mass and the perceived mass. Depending on the timing of the hits, the mass modifier can have three categorical values: a hit around $t_1$ leads to a negative mass modifier ($v_v < v_p$), a hit at $t_2$ ($v_v = v_p$) makes the mass modifier to be zero and for a hit around $t_3$ ($v_v > v_p$), the mass modifier is positive. (D-F) Left panels show the average velocity profile of the hand and the paddle during the last five minutes of adaptation for three individual subjects, one from each possible outcome category. The vertical dashed line represents the average time of the hit in the pong game. Right panels show the adaptation effects on the reaching movements. Error bars represent one standard error of the mean. These results are consistent with the hypothesis that mass estimation occurs at discrete events. (G) Effective mass of the manipulandum in each direction. Each plot is centered on the average position of the hits for the corresponding group. Subjects in the LP$_R$ group played with a heavier paddle than the LP$_L$ group. In addition, the mass modifier is proportional to the mass of the paddle itself. Gray areas represent 95% confidence intervals.

DOI: https://doi.org/10.7554/eLife.32587.004

collision with the ball and the average time of the peak velocity of the paddle during adaptation trials (*Figure 3B*). A paired t-test between the first and the last five minutes of the delayed pong, reveled a significant reduction in the time difference and thereby a progression towards the energy optimal movements ($p = 0.02$). Moreover, during adaptation, the difference between the time of collision with the ball and time of the peak velocity of the paddle was reduced by $0.018 \pm 0.04\, s$ for the LP$_R$ group and $0.019 \pm 0.01\, s$ for the LP$_L$ group. There was not any difference between the two groups in moving towards energy optimal policy (t-test, $p = 0.89$).

In adaptation trials, the force impulse at time of the collision with the ball and auditory feedback were also delayed. Therefore, each hit in the delayed game generated a simultaneous multisensory response similar to the baseline trials. However, participants effectively played the game with two paddles that were separated in time: a visual paddle (delayed) and a proprioceptive paddle (not delayed). *Figure 3C* illustrates the schematic velocity profile of the visual and the proprioceptive paddles for a movement in the hitting direction. For a hit that is happening at time $t_1$ in this figure, the kinetic energy at collision is related to the velocity of the proprioceptive and visual paddles as

$$KE(t_1) = \tfrac{1}{2}m_r v_v(t_1)^2 = \tfrac{1}{2}m_r v_p(t_1 - \tau)^2 \neq \tfrac{1}{2}m_r v_p(t_1)^2 \qquad (1)$$

Where $m_r$, $v_v$, $v_p$ and $\tau$ represent the effective mass of the robot, the velocity of the visual paddle, the velocity of the proprioceptive paddle and the delay, respectively. Also, note that at time $t_1$ the estimate of the kinetic energy of the paddle based on visual information would be different from the estimate based on proprioceptive information because the velocity measurements are different in the two modalities. We hypothesize that sensory integration for mass estimation does not happen continuously in time but only at salient multisensory events when there is an exchange of kinetic energy with the environment, in this case at collisions. Moreover, we hypothesize that the optimization problem must satisfy the constraint that the estimated kinetic energy transfer remains invariant across modalities. Therefore, instead of estimating $\tau$ one may rewrite *Equation (1)* without explicit consideration of the delay, by modifying the effective proprioceptive mass of the robot (see the Materials and methods section for the definition of effective mass and its connection to kinetic energy):

$$KE(t_1) = \tfrac{1}{2}m_r v_v(t_1)^2 = \tfrac{1}{2}(m_r + \hat{m})\, v_p(t_1)^2 \qquad (2)$$

Under this hypothesis, discrete sensory integration at isolated collision events leads to a perceptual illusory mass $\hat{m}$, that hereinafter we refer to as "mass modifier" and can be derived from *Equation (2)* at any hitting time:

$$\hat{m} = m_r \frac{v_v(t_{hit})^2 - v_p(t_{hit})^2}{v_p(t_{hit})^2} \qquad (3)$$

Depending on the time of collision, the mass modifier can have three categorical values (*Figure 3C*): a hit that happens around $t_1$ leads to a negative mass modifier since around this time

$v_v < v_p$, a hit at $t_2$ ($v_v = v_p$) makes the mass modifier to be zero and finally if the hit happens around $t_3$ ($v_v > v_p$), the mass modifier would be positive.

Previously, we examined the changes in the reaching movements following adaptation at the group level. As it is typically the case, there was a substantial variability in the performance of each participant following adaptation. The hypothesis that estimation of the effective mass depends on the sensory measurements at contacts allows us to make predictions of individual responses. To test this prediction, we consider within each group cases that deviated maximally from the average behavior. *Figure 3D* corresponds to the subject that exhibited the largest hypermetria in the $LP_R$ group (right panel). Our hypothesis predicts that the timing of the collisions for this individual should be around $t_3$ because the large hypermetria indicates a positive estimation of the mass modifier and thereby an increase in the perception of the robot's effective mass. Analysis on the pong data confirmed this prediction: the left panel of this figure shows the average velocity profile of the two paddles during the last five minutes of adaptation for this subject and the vertical dashed line represents the average time of the hit. On the other extreme of the $LP_R$ group, the individual in *Figure 3E* did not show an effect. Similar analysis on the pong data showed that on average this subject hit the ball at $t_2$ where the two velocities were equal. Per our hypothesis this would cause the mass modifier to be zero. The subject that exhibited the largest hypermetria in the $LP_L$ group behaved similarly as their counterpart in the $LP_R$ group by timing the strokes in a same manner to hit the ball at around $t_3$ (same as *Figure 3D*). Finally, the other extreme subject in the $LP_L$ group showed a notable hypometria on the right side (*Figure 3F*). In this case, the hypothesis predicts a negative estimation of the mass modifier which is a consequence of the impacts that are occurring at around $t_1$. Subsequent analysis of the velocity profiles and the average hitting time of this subject corroborated with this prediction as well. The three possible outcome categories were not evenly distributed among participants. Most subjects exhibited a significant hypermetria after adaptation (n = 7 in the $LP_R$ group and n = 4 in the $LP_L$ group), while the remaining subjects (n = 1 in the $LP_R$ group and n = 4 in the $LP_L$ group) exhibited either hypermetria or hypometria after adaptation but the confidence intervals of the change after adaptation included zero.

Thus far, we showed that sensory integration at events explains individual differences in all the three possible outcome categories. Later, we will use this concept to model the outcome behavior for all the subjects to further test the hypothesis.

## Mass of the manipulated object explains group differences

The lateral groups played the game in the same direction with the same amount of delay and there were no differences in performance and adaptation rate between the two groups. However, despite the equivalence of the task in the right and left courts, the effect of adaption on the reaching trajectories was asymmetric between these two groups at the end of the experiment. This asymmetry is explained by the change in the dynamics of the task. The effective endpoint mass of the five-bar linkage robotic device used in this study depends on the configuration and the direction of motion. These dependencies can be portrayed by polar plots that are centered at any desired configuration. Each point on the plot represents the projection of the inertia matrix onto the direction (unit velocity vector) that connects the center to that point. *Figure 3G* illustrates two of these plots that are centered on the average position of the hits with arrows that indicate the average movement direction across all subjects in each lateral pong group. This analysis reveals that the subjects in the $LP_R$ group played with an apparently heavier paddle with the effective mass of 1.5 kg, compared to the $LP_L$ group, whose paddle had the average effective mass of 1 kg. We know from *Equation (3)* that the mass modifier is directly proportional to the mass of the object being manipulated and therefore the larger hypermetria in the $LP_R$ group can be explained by the fact that this group played with a paddle that had a larger effective mass than the $LP_L$ group.

## Model predictions

In the previous subsections, we laid out the elements that explain different outcomes at an individual and group level. Here, we present and validate a computational model that employs these concepts to predict the reaching behavior (see the Materials and methods section for a detailed description of the model). For each individual, we extracted the configuration of the robot, velocity of the visual paddle, and velocity of the proprioceptive paddle (hand's velocity) at impacts from the pong data.

From these data, we computed the visual effective mass and the proprioceptive effective mass at hits and combined them by using maximum-likelihood estimation to obtain the mass modifier. Next, we predicted the outcome of blind reaching movements after pong. To this end, we added the mass modifier to the simulated model of the robot and computed the inverse dynamics for pre-planned paths to the targets. We then used the calculated torques as feedforward commands to the actual model of the robot (without the mass modifier) to replicate the blind reaching scenario.

As mentioned earlier, the computational model consists of two main components that collectively explain the results; the timing of the movements and the mass of the object being manipulated. First, we argued that the timing behavior explains individual differences in all three possible outcome categories. Next, we showed that the mass of the object scales the effect, and this explains the asymmetric results between the two group averages in the lateral pong experiments. *Figure 4A* shows the contribution of each component to the total variance. The scatter plots illustrate the average hypermetria for all the subjects in the lateral pong groups on both reaching sides that are sorted based on the magnitude of the hypermetria. If we only consider the mass of the robot, then we have assumed that all the participants in each group learned a single mass modifier that can be most accurately described using the group averages (solid black lines). However, this component only captures 28% of the total variance in the data. On the other hand, if we consider the individual timing behavior in addition to the mass of the robot then we get predictions that explain 61% of the total variance (solid purple lines). The remaining unaccounted variance might be attributable to a multitude of factors including errors in estimating the mechanical parameters and lack of consideration of joint friction of the robot, the fact that the game of pong is a relatively more unconstrained task and that blind reaching movements are inherently nosier than visually guided movements.

*Figure 4B* illustrates the hypermetria in the model predictions averaged across all subject in all the three groups. These predictions demonstrate the ability of this simple computational model with only one free parameter (the mass modifier) to capture the variance in the data: it explains between subject differences, the differences in the magnitudes of the hypermetria across groups and the reduction of the overshoot in the $LP_R$ group on the left side.

## Alternative explanations

There were other potential accounts for some of our observations in this study. However, neither of which was sufficient to explain all the aspects of the results.

1-Proprioceptive recalibration: introducing a visual delay causes a mismatch between vision and proprioception. As we discussed earlier, in the game of pong, the mismatch between the two sensory measurements integrates to zero over time (*Figure 1B*). But if we assume that the recalibration is not occurring continuously and it is limited to the collision events, then, the hypermetria observed in the reaching movements in the frontal pong experiment could also be interpreted as spatial remapping of proprioception. However, in the lateral pong experiments, the direction of the pong was orthogonal to the direction of the reaching movements. In this case, and in contrast to the results, the proprioceptive recalibration model predicts a lateral shift rather than hypermetria in the reaching movements. Alternatively, we can further assume that the direction of the proprioceptive shift also depends on the direction of the movement. To investigate this possibility, we extracted the average spatial mismatch between vision and proprioception at the time of the hits during the last five minutes of adaptation for all the participants of the lateral pong experiments. There was no correlation between the magnitude of the sensory mismatch in pong and the magnitude of the hypermetria in the reaching movements (*Figure 4C*).

2-Visuomotor gain: another possibility is to interpret the results by considering the spatial effect of the imposed delay as a gain factor. A successful ball strike requires the paddle to be at a desired position within a certain time window. To achieve this objective in a delayed visual space, the hand needs to travel a longer distance, in the same time, and in the same direction. Therefore, the spatial distortions brought about by a visual delay can be approximated using a visuomotor scaling factor (*Pine et al., 1996*; *Krakauer et al., 2000*). Indeed, our results from a previous study (*Avraham et al., 2017*) suggested that a visuomotor gain is successful in explaining the transfer of adaptation to a variety of blind movements including reaching and tracking compared to the alternative hypotheses of adaptation in the time domain, a visuomotor shift, and a mechanical system composed of a mass, damper, and spring. Although with a fixed delay, the spatial expansion of the proprioceptive space is not uniform and the scaling factor depends on movement speed, it is

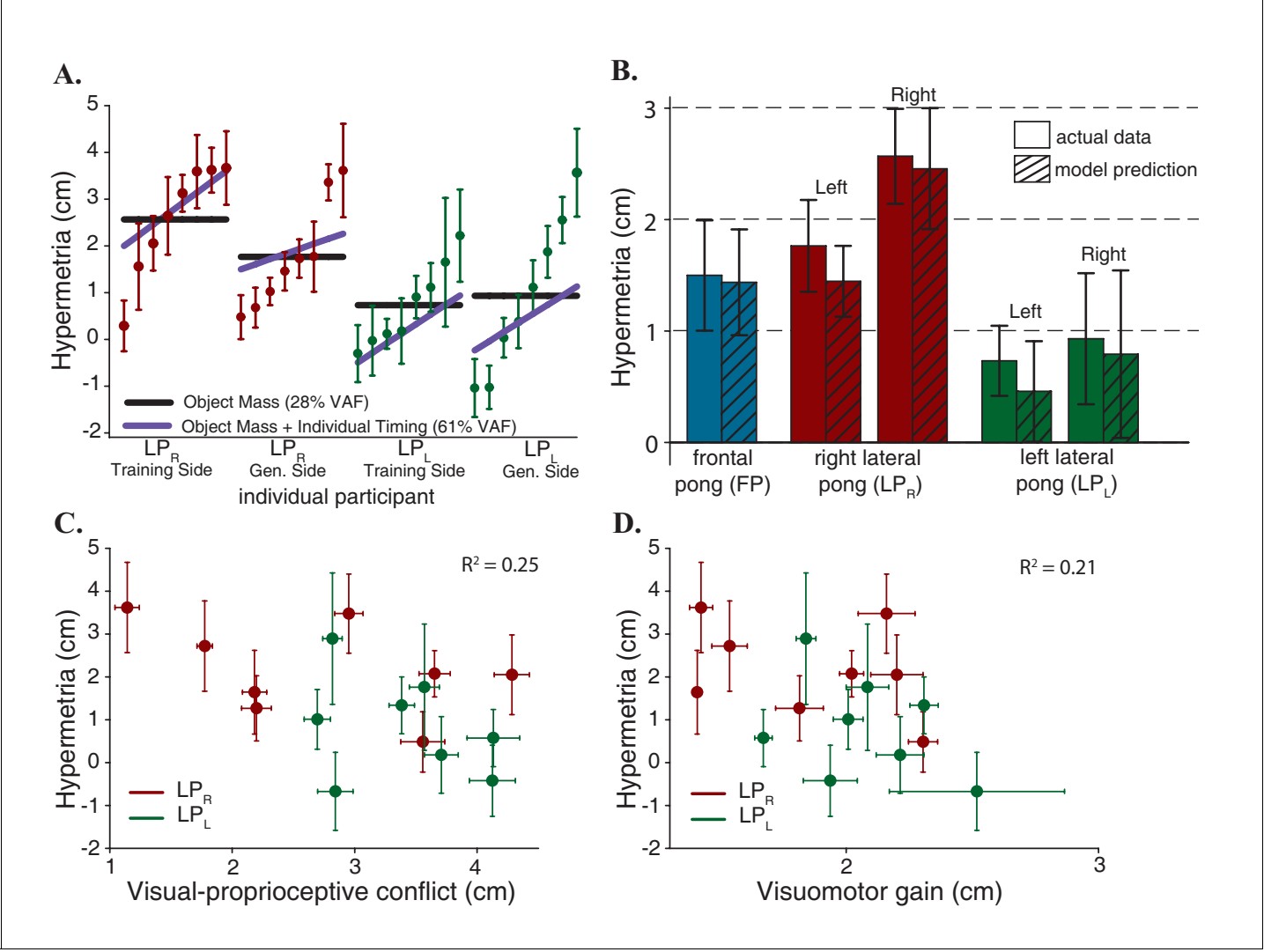

**Figure 4.** Model predictions. (A) Contribution of each component of the model. The scatter plots show the average hypermetria for all the subjects in lateral pong groups on both sides that are sorted based on the magnitude of the hypermetria. Error bars represent 95% confidence interval. Solid black lines represent predictions based solely on the changes in the object mass. Solid purple lines represent predictions based on the changes in the object mass and the individual timing behavior. (B) Model Predictions for all subjects in all three experimental groups. (C) Hypermetria in the reaching movements as a function of average mismatch between sensory measurements at the time of the hits. The absence of correlation indicates that the hypermetria is not caused by proprioceptive recalibration. Error bars represent 95% confidence interval. (D) Hypermetria in the reaching movements as a function of average visuomotor gain experienced during the pong game. The absence of correlation indicates that the hypermetria is not due to adaptation to a scaling perturbation. Error bars represent 95% confidence interval.
DOI: https://doi.org/10.7554/eLife.32587.005

reasonable to assume that the participants learned the average of the scaling factors that they experienced (*Scheidt et al., 2001*; *Braun et al., 2009*). To investigate this possibility, we extracted the average visuomotor gain that participants of the lateral pong groups experienced during the last five minutes of adaptation. The gain factor for each hit was computed as a ratio between the traveled distance of the arm and the paddle from movement initiation to contact. Subsequent analysis also uncovered that there was no correlation between the gain factor during adaptation and the hypermetria in the reaching movements (*Figure 4D*).

3-Mass overestimation due to the visual feedback delay: introducing artificial delays between an applied force and the resulting motion causes an increase in the apparent mass of an object, as it alters the action-consequence relationship (*Honda et al., 2013*). Modeling studies have suggested that in the sensorimotor control system, externally imposed visual delays in the causal link between

force and motion may be approximated by equivalent mechanical systems (*Takamuku and Gomi, 2015*) such as a mass-spring-damper system (*Sarlegna et al., 2010*). Therefore, an alternative explanation is that here as well the effect is due to an excessive delay in the visual response. But an important observation in these and other delay adaptation studies is that the overestimation of the mass fades with adaptation (*Botzer and Karniel, 2013*; *Honda et al., 2013*) and sudden delays in the visual feedback are necessary for the perception of additional mass (*Takamuku and Gomi, 2015*). On the contrary, here the effect is a consequence of prolonged exposure and adaptation.

Furthermore, the asymmetric results from the lateral pong experiments allowed us to further reject these alternative possibilities. Two groups of participants played the game with the same temporal delay and identical kinematics while holding simulated paddles with different inertial mass. After adaptation, they exhibited a significantly different pattern of reaching movements. This asymmetric outcome eliminates the entire class of kinematic models. The free parameters in the mechanical equivalent model are also derived using the kinematics (position, velocity, and acceleration) of the object and its delayed representation. Therefore, this model also predicts an equal additional mass to be perceived by the groups. Moreover, none of these models could account for individual differences among the participants.

## Discussion

We examined the temporal structure of the estimation process that is involved in the representation of object dynamics. Participants played a virtual pong game under an artificially induced visuomotor delay and performed reaching movements (without visual feedback) before and after the game using a robotic manipulandum. We predicted that a continuous or periodic estimation should result in no change in the internal representation of the robot's inertia whereas discrete estimation at contact events should lead to changes in the represented inertia. The existence of neural representations, called 'internal models', of the physical properties of the limbs and of the objects being manipulated is a widely accepted view that has nevertheless attracted some controversy. Feldman and Latash (*Feldman and Latash, 2005*) have rejected altogether the idea that the brain 'computes' mechanical variables such as force, torque and inertia in a way similar to how robotic manipulators are programmed. Instead, they posit that muscle mechanical properties combined with the modulation of reflex parameters may render the anticipation of the mechanical consequences of motor commands unnecessary. While we instead remain convinced that the long delays of neural communications - both sensory and motor - and the complex nonlinear mechanics of the limbs and environment provide a strong rationale for predictive internal representations, we regard the existence and character of such representations far from being settled. We would not suggest that neural computation follows the same structure as robot control programs. In fact, the development of adaptive internal models based on observed statistics of commands and outcome is not a priority for robotic systems that can operate at feedback rates that are order of magnitude faster than the biological counterparts. We consider internal models as a conceptual tool to organize hypotheses and experiments on the brain's ability to plan actions and form expectations based on the assessed balance between deterministic and random influences of the environment. In this spirit, we found changes in the reaching trajectories after the game suggesting that participants estimated the mass only during contact events, at which kinetic energy was exchanged with the environment. These modifications in mass estimations appeared to conserve the expected exchange of kinetic energy across sensing modalities.

### Multisensory events in object manipulation

Many discrete events in the physical world are perceived through multiple sensory modalities providing us with different types of information regarding those events. Although these sensory stimuli originate synchronously in the environment, to perceive them as simultaneous the nervous system should, and in fact does, account for the differences in both physical (*Sugita and Suzuki, 2003*) and neural (*Stone et al., 2001*) transmission rates. Additionally, the neural mechanism of simultaneity perception is subjective (*Vroomen and Keetels, 2010*) and the temporal intermodal alignment can be recalibrated (*Fujisaki et al., 2004*). This adaptive mechanism is proposed to be beneficial for object manipulation purposes (*Johansson and Flanagan, 2009*). Manipulation tasks often include distinct action phases in which objects are grasped, moved, brought in contact with other objects

and released. These action phases are confined between discrete contact events that generate multi-sensory responses which are linked in space and time (*Flanagan et al., 2006*; *Johansson and Flanagan, 2009*). Therefore, sensory integration at these events provides a more accurate and reliable perception of the environment. In this study, we have provided experimental evidence to suggest that the nervous system exploits this opportunity by limiting the context estimation to sensory information provided at multimodal events.

When reaching to grasp objects, the brain predicts the sensory consequences of contacts and estimate the level of the required grip force before they happen (*Flanagan and Beltzner, 2000*; *Flanagan et al., 2003*) using the experience of the previously manipulated objects (*Haruno et al., 2001*). Contact events are rich sources of information to compare the predicted and actual sensory responses. Therefore, forward models can be updated and aligned using prediction errors and context estimation at events. Depending on the complexity of the interactions and past experiences, occasional regulation of the forward models at events could be sufficient to fulfill and attain the manipulation objectives. Indeed, this was the observation in the current study. Research on eye-hand coordination in sequential object manipulation tasks reported that participants direct their gaze to successive contact locations that mark the end of a sequence well before the time that hand reaches them (*Johansson et al., 2001*; *Flanagan and Johansson, 2003*). But the gaze location remains fixed and stationary until the sequence is completed. These results indicate that the sensorimotor control system is actively seeking for task-relevant events that provide distinct and simultaneous multi-sensory information to compare and regulate forward model predictions for the upcoming manipulation sequence, while being confident that the previous event-based adjustments were adequate to attain the objective of the current sequence without any additional use for feedback. This event-driven use of state feedback in sensorimotor control has obvious computational advantages over a control scheme that continuously incorporates feedback.

## Generalization

We put forward that the event-driven employment of feedback for context estimation and forward models' calibration is not limited to contacts. External perturbations and inaccurate forward models lead to performance and prediction errors that require correction. Feedback is then integrated only after an event indicates that the control error exceeded some threshold. This threshold is variable and depends on feedback uncertainty (*Wei and Körding, 2010*), perturbation uncertainty (*Izawa et al., 2008*) and the level of precision that is required by the task itself. Therefore, similar to the contact events, error events adjust forward models only in task-relevant dimensions. This task dependent use of feedback allows forward models to drift in the task-irrelevant dimensions (uncontrolled manifold) over time (*Scholz and Schöner, 1999*; *Todorov and Jordan, 2002*). In novel object manipulation tasks, when there are no forward models to rely on, feedback is extensively utilized at initial stages to train forward models whereas practice reduces reliance on feedback (*Sailer et al., 2005*).

We propose that the features that we discussed so far regarding context estimation can be generalized to state estimation. Ariff and colleagues (*Ariff et al., 2002*) designed an experiment in which they asked participants to track with their eyes the location of their own unseen hand during reaching movements and they found a proactive gaze behavior with gaze leading the hand. In this task, forwards models and proprioceptive feedback were combined to estimate the state of the hand and eye movements were served as a proxy for the estimation process. An important observation in this study - for our purposes here - is that rather than pursuit eye movements, participants made saccades to track the hand (but see [*Gauthier and Mussa Ivaldi, 1988*; *Gauthier et al., 1988*]). Moreover, the position and timing of these saccades were random. Therefore, even in simple and familiar reaching movements, the task demands for continuous state estimation could not be satisfied.

## The role of cerebellum in event prediction and formation of forward models

The adaptive learning mechanism in the cerebellum (*Marr and Thach, 1991*) makes it an ideal substrate for generating forward models. There is growing body of evidence from studies on behavioral deficits in patients with cerebellar dysfunction (*Bastian et al., 1996*; *Tseng et al., 2007*), functional imaging (*Blakemore et al., 2001*; *Kawato et al., 2003*), and transcranial magnetic stimulation

(*Miall et al., 2007*; *Schlerf et al., 2012*) that links the cerebellum to forward models (*Bastian, 2006*). In a ball catching task, subjects with cerebellar damage exhibited difficulty in predicting the required muscle forces to compensate for ball weight before the ball reached the hand, but showed normal force adjustments after impact (*Lang and Bastian, 1999*). Similarly, in a locomotion study (*Morton and Bastian, 2006*), Subjects with cerebellar damage were capable of making reactive changes to a perturbation, but were impaired at making predictive adjustments. In object manipulation, cerebellar lesions prevented predictive grip force modulations in anticipation of inertial loads (*Nowak et al., 2002*; *Rost et al., 2005*). These results suggest that the integrity of the cerebellum is critical for preparing motor responses in anticipation of discrete sensory events that mark the transition between action phases. Damages to the cerebellum impairs adaptation to both kinematic (*Martin et al., 1996*) and dynamic (*Smith and Shadmehr, 2005*) changes in the environment. Persons with cerebellar ataxia may exhibit dysmetria in their movements. The dysmetria have a distinctive character in each individual. Some tend to show hypometria, while others are hypermetric (*Manto, 2009*). It has recently been shown that errors in movement extent in patients with cerebellar dysmetria is caused by the misrepresentation of arm dynamics (*Bhanpuri et al., 2014*). Our findings here suggest that errors in estimating mechanical properties of the arm could be caused by the cerebellar dysfunction in temporal processing and alignment of multimodal sensory information. Moreover, each injury to the cerebellum, depending on the location and severity, leads to a specific temporal calibration error in sensory integration causing a broad range of patient specific motor deficits.

## Materials and methods

### Participants

Twenty-four right handed volunteers (11 females, ranging in age from 23 to 35) participated in the study. All participants were neurologically intact with normal or corrected to normal vision and had no prior knowledge of the experimental procedure. The study protocol was approved by Northwestern University's Institutional Review Board (STU00026226) and all the participants signed an informed consent form.

### Experimental setup

Participants were positioned in front of a horizontal screen and held the handle of a planar, two-degree of freedom robotic manipulandum with their right hand. The screen prevented the participant's view of their arm and the robot. A projector was used to display the visual information on the screen and it was calibrated so that the position of the handle was overlaid on its true position with a precision of 1 mm. Position and velocity of the robot were computed from instrumented encoders at the frequency of 1 kHz to provide sensory feedback during the experiment and the data were recorded at the rate of 200 Hz.

### Experimental design

The experiment consisted of two tasks: playing a pong game and executing reaching movements. In the pong game, the ball movement was confined to a rectangular court and participants were instructed to hit the ball towards a side that was distinguished by a different color (green sides in *Figure 1A* and *Figure 2A*) from the remaining sides using a rectangular paddle that represented the location of the hand. To expand the court coverage and to mimic the presence of an opponent, the velocity of the ball was changed by a random number upon bouncing from the distinguished side of the court. This number was drawn from a uniform distribution between $\pm 0.13$ $m/s$ and was applied to the velocity component along the bouncing side. Additionally, friction was modeled as a linear decay in the velocity of the ball. After a collision with the paddle, the ball velocity was determined using

$$\begin{bmatrix} \dot{x}_{ball}^+ \\ \dot{y}_{ball}^+ \end{bmatrix} = 0.7 \begin{bmatrix} \cos 2\theta & \sin 2\theta \\ \sin 2\theta & -\cos 2\theta \end{bmatrix} \begin{bmatrix} \dot{x}_{ball}^- \\ \dot{y}_{ball}^- \end{bmatrix} + 0.42 \begin{bmatrix} \dot{x}_{paddle} \\ \dot{y}_{paddle} \end{bmatrix} \qquad (4)$$

Where − and + represent before and after the collision respectively and θ is the orientation of the paddle with respect to the horizontal axis. A haptic pulse was generated by the robot at the time of impact for the duration of 5 $ms$. This force feedback was computed using $f = m_b \, \Delta v_{ball}$, where $m_b =$

$0.05\,kg$ and $\Delta v$ is the change in the velocity vector. The sudden activation of the motors to generate this pulse was creating a sound that made it unnecessary to provide any additional auditory feedback. Each trial of the pong game lasted for one minute. A timer indicated the elapsed time and a counter displayed the number of collisions in each trial.

In the reaching phase, the screen turned black and a circular target appeared on the screen. Participants were instructed to reach the target and stop there. This movement was executed without a visual feedback of the location of the hand and it was guided only by the proprioceptive representation of the hand position in relation to the visual target. After the movement was complete, the hand was passively brought back to the starting position by the robot. Similarly, no visual feedback of the starting position was present.

## Protocol

Participants were randomly divided in three groups. All the experiments consisted of a reaching-pong-reaching sequence. Participants in the first experiment (n = 8), played pong in frontal direction (**Figure 1A**). After two minutes, the game was delayed for $\tau = 80\,ms$ and participants played the delayed game for ~40 minutes. The delay was applied across all the visual, haptic, and auditory channels. In reaching tasks, participants performed 45 reaching movements in a random order to three targets that were placed at $0.14\,m$ from the starting position and were separated from each other by $45°$ (**Figure 1A**). A subgroup of participants in this experiment (n = 5), also participated in a control experiment in a separate session where they played the game for ~20 minutes but without the pong being delayed. The order in which these participants performed the delayed and non-delayed game was randomized.

In the two other experiments, participants played pong in lateral direction. Two pong courts where juxtaposed next to each other (**Figure 2A**) in such a way that their intersection was along the participants' body midline. Each court was a square with the side of $0.2\,m$ and subjects made contacts with the ball on average in the middle of the court. Therefore, they had to move $0.1\,m$ in the lateral direction from the midline of the body. This movement was by far within the area that each subject could comfortably reach. At the beginning, participants played the pong game with no delay in both courts for the total time of eight minutes that was equally divided and alternating between the courts. Next, participants in one group (n = 8) played the delayed pong only in the right court for 40 min, while the other group (n = 8) played the delayed pong only in the left court for the same amount of time and with the same amount of delay ($\tau = 120\,ms$). However, the reaching tasks before and after pong were identical across these two groups. In the lateral pong experiments the direction of the reaching was orthogonal to the direction of the pong. We duplicated the same pattern of targets that was used in the first experiment and placed one in each court (**Figure 2A**). Therefore, participants in these two groups performed reaching movements to six targets (three in each court) from two corresponding starting positions (one in each court). Each movement was repeated five times in a random order.

## Computational model

The equations of motion for a five-bar linkage robotic device used in this experiment can be derived using the Euler-Lagrange equations and expressed in matrix form as

$$M(q)\ddot{\boldsymbol{q}} + C(\boldsymbol{q},\dot{\boldsymbol{q}})\dot{\boldsymbol{q}} = u \tag{5}$$

Where $M(q)$, $C(q,\dot{q})$ and $u$ represent the inertia matrix, Centripetal/Coriolis matrix and generalized forces respectively. The effective mass of the robot is spatially varying and configuration dependent. The effective mass ($m_r$) is defined as the projection of the inertia matrix onto the instantaneous direction of motion (**Worsnopp et al., 2006**). Therefore, for each direction of motion at each configuration, the inertia of the robot matches that of a point mass. The inertia of this equivalent point mass can be derived using the conservation of energy principle: the kinetic energy of the robot must be equivalent to the kinetic energy of the point mass

$$\frac{1}{2}\dot{\boldsymbol{q}}^T M(\boldsymbol{q})\dot{\boldsymbol{q}} = \frac{1}{2}\dot{\boldsymbol{x}}^T m_r \dot{\boldsymbol{x}} \tag{6}$$

The unit velocity vector is $\dot{x} = [\cos\theta\ \sin\theta]^T$ and $\theta$ is the angle between the direction of motion and the x-axis. Therefore

$$m_r = \dot{x}^T \left(J^{-1}(q)\right)^T M(q) J^{-1}(q) \dot{x} \tag{7}$$

Where $J$ is the Jacobian matrix. To predict the outcome of blind reaching movements after pong we extracted the configuration of the robot, the velocity of hand ($v_p$) and its delayed representation ($v_v$) at hits from the pong data during the last five minutes of adaption for each individual. From these data, we computed the effective visual mass ($m_v = m_r$) and the effective proprioceptive mass ($m_p = m_r \frac{v_v^2}{V_p^2}$) of the robot. Next, we integrated these sensory information using maximum-likelihood estimation (Ernst and Banks 2002) to obtain the apparent mass

$$E(m) = \frac{\frac{1}{\sigma_v^2}}{\frac{1}{\sigma_v^2} + \frac{1}{\sigma_p^2}} m_v + \frac{\frac{1}{\sigma_p^2}}{\frac{1}{\sigma_v^2} + \frac{1}{\sigma_p^2}} m_p \tag{8}$$

Where $\sigma_v^2$ and $\sigma_p^2$ represent the variance of the effective visual and proprioceptive masses at hits. The perceived mass is therefore different from the actual effective mass of the robot. We called this difference the mass modifier ($\hat{m} = E(m) - m_r$).

Finally, we added the mass modifier to the simulated model of the robot and computed the inverse dynamics for preplanned minimum jerk (*Flash and Hogan, 1985*) trajectories to the targets. We then used the calculated torques as feedforward commands to the actual model of the robot (without the mass modifier). The difference in the dynamic model of the robot between inverse computation and feedforward simulation caused an erroneous trajectory. The magnitude of the error was used to emulate changes in reaching trajectories after adaptation.

## Data and statistical analysis

A fifth-order Butterworth low-pass filter with a cutoff frequency of 20 Hz was implemented to smooth the velocity signals. We fed the hit data from the last five minutes of pong to the computational model, however the output of the model was not sensitive to this choice. The hits at which the proprioceptive effective mass was more than ten standard deviations away from the mean visual effective mass were removed from the analysis. The threshold of significance in all the statistical analysis was set at 0.05.

## Acknowledgement

This material is based upon work supported by the National Science Foundation under Grant No. 1632259 awarded to FAMI. The Binational United-States Israel Science Foundation (grants no. 2011066, 2016850) awarded to FAMI and IN, the Israel Science Foundation (grant no. 823/15), and the Helmsley Charitable Trust through the Agricultural, Biological and Cognitive Robotics Initiative of Ben-Gurion University of the Negev, Israel, both awarded to IN. We thank A Karniel, R Shadmehr, R Scheidt, L Simo, E Perreault, T Murphey, R Ranganathan, F Huang, and E Thorp for discussions and comments.

## Additional information

### Funding

| Funder | Grant reference number | Author |
| --- | --- | --- |
| National Science Foundation | 1632259 | Ferdinando A Mussa-Ivaldi |

The funders had no role in study design, data collection, and interpretation, or the decision to submit the work for publication.

## Author contributions

Ali Farshchian, Conceptualization, Resources, Data curation, Software, Formal analysis, Validation, Investigation, Visualization, Methodology, Writing—original draft, Writing—review and editing; Alessandra Sciutti, Conceptualization, Resources, Data curation, Software, Formal analysis, Validation, Investigation, Visualization, Methodology, Writing—review and editing; Assaf Pressman, Conceptualization, Data curation, Investigation, Methodology, Writing—review and editing; Ilana Nisky, Conceptualization, Supervision, Writing—review and editing; Ferdinando A Mussa-Ivaldi, Conceptualization, Supervision, Funding acquisition, Writing—original draft, Writing—review and editing

## Author ORCIDs

Ali Farshchian (ID) http://orcid.org/0000-0001-9321-0944
Alessandra Sciutti (ID) http://orcid.org/0000-0002-1056-3398
Ilana Nisky (ID) https://orcid.org/0000-0003-4128-9771
Ferdinando A Mussa-Ivaldi (ID) http://orcid.org/0000-0001-5343-7052

## Ethics

Human subjects: The study protocol was approved by Northwestern University's Institutional Review Board (STU00026226) and all the participants signed an informed consent form.

## Decision letter and Author response

Decision letter https://doi.org/10.7554/eLife.32587.010
Author response https://doi.org/10.7554/eLife.32587.011

# Additional files

## Supplementary files

• Transparent reporting form
DOI: https://doi.org/10.7554/eLife.32587.006

## Data availability

Data files for this manuscript are available through Dryad doi:10.5061/dryad.93kc5cb

The following dataset was generated:

| Author(s) | Year | Dataset title | Dataset URL | Database, license, and accessibility information |
|---|---|---|---|---|
| Farshchian A, Sciutti A, Pressman A, Nisky I, Mussa-Ivaldi S | 2018 | Data from: Energy exchanges at contact events guide sensorimotor integration across intermodal delays | http://dx.doi.org/10.5061/dryad.93kc5cb | Available at Dryad Digital Repository under a CC0 Public Domain Dedication |

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
