## [Decision Letter]

Thank you for submitting your article "Energy Exchanges at Contact Events Guide Sensorimotor Integration Across Intermodal Delays" for consideration by *eLife*.

Your article has been reviewed by two peer reviewers, and I have also reviewed the paper, serving in the dual role of Reviewing and Senior Editor. One of the reviewers has agreed to reveal his name, Joseph McIntyre. We have worked together to provide this decision letter to help you prepare a revised submission.

We all believe you are addressing an interesting problem and found the experimental work sound. The work centers on two core hypotheses: (1) Adaptation to artificially delayed feedback is based on the principle of equivalent energy exchange and adaptation of the estimated mass used for prediction, and (2) The adaptive process does not sample information continuously, but only at discrete events, which in this case are the collision events. Based on an extended discussion between the reviewers and myself, we would like the revision to address the following issues:

1) If one accepts #1, the experiments provide good support for #2. However, we think there should be an extended discussion of alternatives. You lay out two alternative ideas in the General Discussion, one centered on the idea that adaptation affects the perceived visual mass and the other about a scaling factor (gain). While you offer some arguments against these ideas, we think it would be more convincing if you provided a more complete analysis of the adaptation patterns. We also considered two additional alternative hypotheses:

A) Adaptation of spatial mapping between vision and proprioception. We believe that a more comprehensive analysis is required to reject this hypothesis. Specifically, we ask that you perform an analysis of the inter-subject differences in kinematic parameters related to this hypothesis, similar to what you have done with your energy exchange hypothesis.

B) Adaptation in the time domain. As one reviewer noted, if adaptation was to result in a time shift of say, the visual function (Figure 1B), then the continuous and discrete hypotheses would make the same prediction since the functions completely overlap. Of course, by your perspective, the correspondence would mean that there should be no hypermetria. I realize you have done other work that argues against the idea that delayed feedback results in adaptation of time. Moreover, I assume you would also argue from the current results that if there was temporal adaptation resulting in an alignment of the two functions, then there should be no hypermetria (both hypotheses would predict this). We would like to see some explicit discussion of this issue, especially since readers are surely going to consider whether these effects are due to adaptation in time (given work by others who have argued for temporal adaptation using similar tasks (e.g., Fujisaki et al., 2004, Keetels and Vroomen, 2012).

2) We would like to see more information concerning performance changes during the adaptation phase. The present draft is limited to an analysis of hit rates, early and late in the adaptation phase. We would like to see evidence, likely in terms of measures of movement extent and/or duration of what is changing here. As one reviewer wrote, "Perhaps there is no adaptation and subjects overshoot targets because they keep moving until the visual cursor lands in the target zone."

3) Related to point #2, in the lateral condition (Experiment 2 and Experiment 3) you argue that there are no learning differences between the two sides, focusing on an ANOVA of hit rates. Again, would like to see a more rigorous analysis of the adaptation function for the two sides. For this experiment, it is also important to show that there are no systematic pre-adaptation differences in movements made to the two halves of the workspace. This is also important because you want to argue that the asymmetries after training arise because of contact with the pong ball. But given the differences in the robot arm inertial anisotropy at the left and the right, it is possible that these differences impact adaptation (e.g., might occur in absence of pong game).

4) Consider reworking the order of the paper. As currently written, the motivation for Experiments 2/3 (which I think should be just called Experiment 2 with two conditions) is not clear and the explanation for the asymmetric results is not provided until much further in the manuscript. You could consider providing the key theoretical analysis right after Experiment 1 and then use this to motivate Experiment 2 in a more predictive manner ("This model leads to the prediction…"). I'm not endorsing post-hoc science writing here but assume this is what actually motivated Experiment 2. If you feel this reorganization suggestion would not help with the flow of the paper, then feel free to ignore this recommendation.

---

## [Author Response]

[…] The work centers on two core hypotheses: (1) Adaptation to artificially delayed feedback is based on the principle of equivalent energy exchange and adaptation of the estimated mass used for prediction, and (2) The adaptive process does not sample information continuously, but only at discrete events, which in this case are the collision events. Based on an extended discussion between the reviewers and myself, we would like the revision to address the following issues:1) If one accepts #1, the experiments provide good support for #2. However, we think there should be an extended discussion of alternatives. You lay out two alternative ideas in the General Discussion, one centered on the idea that adaptation affects the perceived visual mass and the other about a scaling factor (gain). While you offer some arguments against these ideas, we think it would be more convincing if you provided a more complete analysis of the adaptation patterns. We also considered two additional alternative hypotheses:A) Adaptation of spatial mapping between vision and proprioception. We believe that a more comprehensive analysis is required to reject this hypothesis. Specifically, we ask that you perform an analysis of the inter-subject differences in kinematic parameters related to this hypothesis, similar to what you have done with your energy exchange hypothesis.B) Adaptation in the time domain. As one reviewer noted, if adaptation was to result in a time shift of say, the visual function (Figure 1B), then the continuous and discrete hypotheses would make the same prediction since the functions completely overlap. Of course, by your perspective, the correspondence would mean that there should be no hypermetria. I realize you have done other work that argues against the idea that delayed feedback results in adaptation of time. Moreover, I assume you would also argue from the current results that if there was temporal adaptation resulting in an alignment of the two functions, then there should be no hypermetria (both hypotheses would predict this). We would like to see some explicit discussion of this issue, especially since readers are surely going to consider whether these effects are due to adaptation in time (given work by others who have argued for temporal adaptation using similar tasks (e.g., Fujisaki et al., 2004, Keetels and Vroomen, 2012).

We have now moved the section of the alternative models from the Discussion section to the Results section and now it includes three alternatives:

1) Proprioceptive recalibration: introducing a visual delay causes a mismatch between vision and proprioception. As we discussed earlier, in the game of pong, the mismatch between the two sensory measurements integrates to zero over time (Figure 1B). But if we assume that the recalibration is not occurring continuously, and it is limited to the collision events, then, the hypermetria observed in the reaching movements in the frontal pong experiment could also be interpreted as spatial remapping of proprioception. However, in the lateral pong experiments, the direction of the pong was orthogonal to the direction of the reaching movements. In this case, and in contrast to the results, the proprioceptive recalibration model predicts a lateral shift rather than hypermetria in the reaching movements. Alternatively, we can further assume that the direction of the proprioceptive shift also depends on the direction of the movement. To investigate this possibility, we extracted the average spatial mismatch between vision and proprioception at the time of the hits during the last five minutes of adaptation for all the participants of the lateral pong experiments. This analysis revealed that there was no correlation between the magnitude of the sensory mismatch in pong and the magnitude of the hypermetria in the reaching movements (Figure 4C).

2) Visuomotor gain: another possibility is to interpret the results by considering the spatial effect of the imposed delay as a gain factor. A successful ball strike requires the paddle to be at a desired position within a certain time window. To achieve this objective in a delayed visual space, the hand needs to travel a longer distance, in the same time, and in the same direction. Therefore, the spatial distortions brought about by a visual delay can be approximated using a visuomotor scaling factor (Pine and Krakauer, 1996, Krakauer and Pine, 2000). Indeed, our results from a previous study (Avraham et al., 2017) suggested that a visuomotor gain is successful in explaining the transfer of adaptation to a variety of blind movements including reaching and tracking compared to the alternative hypotheses of adaptation in the time domain, a visuomotor shift, and a mechanical system composed of a mass, damper, and spring. Although with a fixed delay, the spatial expansion of the proprioceptive space is not uniform, and the scaling factor depends on movement speed, it is reasonable to assume that the participants learned the average of the scaling factors that they experienced (Scheidt and Dingwell, 2001, Braun and Aertsen, 2009). To investigate this possibility, we extracted the average visuomotor gain that participants of the lateral pong groups experienced during the last five minutes of adaptation. The gain factor for each hit was computed as a ratio between the traveled distance of the arm and the paddle from movement initiation to contact. This analysis also uncovered that there was no correlation between the gain factor during adaptation and the hypermetria in the reaching movements (Figure 4D).

3) Mass overestimation due to the visual feedback delay: introducing artificial delays between an applied force and the resulting motion causes an increase in the apparent mass of an object, as it alters the action-consequence relationship (Honda and Hagura, 2013). Modeling studies have suggested that in the sensorimotor control system, externally imposed visual delays in the causal link between force and motion may be approximated by equivalent mechanical systems (Takamuku and Gomi, 2015) such as a mass-spring-damper system (Sarlegna and Baud-Bovy, 2010). Therefore, an alternative explanation is that here as well the effect is due to an excessive delay in the visual response. But an important observation in these and other delay adaptation studies is that the overestimation of the mass fades with adaptation (Botzer and Karniel, 2013, Honda and Hagura, 2013) and sudden delays in the visual feedback are necessary for the perception of additional mass (Takamuku and Gomi, 2015). On the contrary, here the effect is a consequence of prolonged exposure and adaptation.

Furthermore, the asymmetric results from the lateral pong experiments allowed us to further reject these alternative possibilities. Two groups of participants played the game with the same temporal delay and identical kinematics while holding simulated paddles with different inertial mass. After adaptation, they exhibited a significantly different pattern of reaching movements. This asymmetric outcome eliminates the entire class of kinematic models. The free parameters in the mechanical equivalent model are also derived using the kinematics (position, velocity, and acceleration) of the object and its delayed representation. Therefore, this model also predicts an equal additional mass to be perceived by the groups. Moreover, none of these models could account for individual differences among the participants.

However, we do not argue against the adaptation in the time domain. In fact, our analysis (please see comment #2) indicates that during adaptation there was a progressive temporal shift towards the energetically optimal movements. The fact that the reaching trajectories were changed after playing the delayed game does not provide evidence against temporal adaptation, but it indicates that sensory integration for mass estimation was limited to the time of the impact and therefore the movement timing of each participant during adaptation was a predictor of this change. For this reason, in the case of complete adaptation (i.e. hitting the ball at the peak velocity of the paddle) the reaches were hypermetric (Figure 3D) while, there was no hypermetria only in the case in which the sensory measurements at the time of the events were equal (Figure 3E).

2) We would like to see more information concerning performance changes during the adaptation phase. The present draft is limited to an analysis of hit rates, early and late in the adaptation phase. We would like to see evidence, likely in terms of measures of movement extent and/or duration of what is changing here.

We have now added the following figure and analysis to the manuscript.

Our analysis on the baseline, non-delayed pong trials revealed that the participants optimized the kinetic energy transfer to the ball by hitting the ball at the time of peak velocity (Figure 3A). Introducing the delay affected this optimal behavior and therefore, the most relevant metric, that also provides support for our line of reasoning here, is the one that could show with adaptation there was a continuous effort towards recovering the energy optimal behavior. We have computed the time difference between the average time of collision with the ball and the average time of the peak velocity of the paddle during adaptation trials (Figure 3B). A paired t-test between the first and the last five minutes of the delayed pong, reveled a significant reduction in the time difference and thereby a progression towards the energy optimal movements (𝑝𝑝 = 0.02).

As one reviewer wrote, "Perhaps there is no adaptation and subjects overshoot targets because they keep moving until the visual cursor lands in the target zone."

To assess the changes in representation of mass, we asked participants to perform reaching movements without any feedback (in a feedforward fashion) before and after playing pong. So, no visual cursor was present at any time during the reaching movements.

3) Related to point #2, in the lateral condition (Experiment 2 and Experiment 3) you argue that there are no learning differences between the two sides, focusing on an ANOVA of hit rates. Again, would like to see a more rigorous analysis of the adaptation function for the two sides.

During adaptation, the difference between the time of collision with the ball and time of the peak velocity of the paddle was reduced by 0.018 ± 0.04s for the LP_R_ group and 0.019 ± 0.01 for the LP_L_ group. There was not any difference between the two groups in moving towards energy optimal policy (t-test, 𝑝𝑝 = 0.89). We have now also compared an additional kinematic feature of the movements during the pong game between two groups in the lateral condition and there was no difference in movement extent (t-test, 𝑝𝑝 = 0.18) between the two groups at the end of the adaptation. We have added this analysis to the manuscript.

For this experiment, it is also important to show that there are no systematic pre-adaptation differences in movements made to the two halves of the workspace. This is also important because you want to argue that the asymmetries after training arise because of contact with the pong ball. But given the differences in the robot arm inertial anisotropy at the left and the right, it is possible that these differences impact adaptation (e.g., might occur in absence of pong game).

To ensure that the difficulty level of playing pong was not different between the courts, initially we asked all participants to play the game with no delay in both courts. Hit rate analysis showed that there was no difference in performance across the courts for each individual (paired t-test, 𝑝𝑝 = 0.32). Thus, we could assume that there was not an inherent gap in difficulty between the two courts. The asymmetries after training was at a group level. In addition, there was no significant difference in the movement extent (t-test, 𝑝𝑝 = 0.5) between the movements made by the LP_R_ group on the right court and the movements made by the LP_L_ group on the left court during the pre-adaptation pong. We have added this analysis to the manuscript.

4) Consider reworking the order of the paper. As currently written, the motivation for Experiments 2/3 (which I think should be just called Experiment 2 with two conditions) is not clear and the explanation for the asymmetric results is not provided until much further in the manuscript. You could consider providing the key theoretical analysis right after Experiment 1 and then use this to motivate Experiment 2 in a more predictive manner ("This model leads to the prediction…"). I'm not endorsing post-hoc science writing here but assume this is what actually motivated Experiment 2. If you feel this reorganization suggestion would not help with the flow of the paper, then feel free to ignore this recommendation.

We have now provided the motivation for the Experiment 2 at the end of the first experiment and stated the hypothesis upfront and explained why asymmetric results are expected for the second experiment.